# Experimental Evaluation of Radiation Response and Thermal Properties of NPs-Loaded Tissues-Mimicking Phantoms

**DOI:** 10.3390/nano12060945

**Published:** 2022-03-13

**Authors:** Somayeh Asadi, Sanzhar Korganbayev, Wujun Xu, Ana Katrina Mapanao, Valerio Voliani, Vesa-Pekka Lehto, Paola Saccomandi

**Affiliations:** 1Department of Mechanical Engineering, Politecnico di Milano, 20156 Milan, Italy; sanzhar.korganbayev@polimi.it (S.K.); paola.saccomandi@polimi.it (P.S.); 2Department of Applied Physics, University of Eastern Finland, 70211 Kuopio, Finland; wujun.xu@uef.fi (W.X.); vesa-pekka.lehto@uef.fi (V.-P.L.); 3Center for Nanotechnology Innovation, Istituto Italiano di Tecnologia, 56127 Pisa, Italy; katrina.mapanao@sns.it (A.K.M.); valerio.voliani@iit.it (V.V.)

**Keywords:** thermal properties, hyperthermia therapy, nanoparticles, irradiation response, tissue-mimicking phantom

## Abstract

Many efforts have recently concentrated on constructing and developing nanoparticles (NPs) as promising thermal agent for optical hyperthermia and photothermal therapy. However, thermal energy transfer in biological tissue is a complex process involving different mechanisms such as conduction, convection, radiation. Therefore, having information about thermal properties of tissue especially when NPs are embedded in is a necessity for predicting the heat transfer during hyperthermia. In this work, the thermal properties of solid phantom based on agar in the presence of three different nanoparticles (BPSi, tNAs, GNRs) and alone were measured and reported as a function of temperature (ranging from 22 to 62 °C). The thermal response of these NPs to an 808 nm laser beam with three different powers were studied in the water comparatively. Agar and tNAs have almost constant thermal properties in the considered range. Among the three NPs, gold has the highest conductivity and diffusivity. At 62 °C BPSi NPs have the similar amount of increase for the diffusivity. The thermal parameters reported in this paper can be useful for the mathematical modeling. Irradiation of the NPs-loaded water phantom displayed the highest radiosensitivity of gold among the three mentioned NPs. However, for the higher power of irradiation, BPSi and tNAs NPs showed the increased absorption of heat during shorter time and the increased temperature gradient slope for the initial 15 s after the irradiation started. The three NPs showed different thermal and irradiation response behavior; however, this comparison study notes the worth of having information about thermal parameters of NPs-loaded tissue for pre-clinical planning.

## 1. Introduction

Tumors in some organs are quite difficult to remove without significant damage to healthy tissue nearby. Despite significant progress that has been achieved in the cancer treatment through thermal-based techniques, there is still a concern associated with the tumor-surrounding normal cells. Laser-based thermotherapy is one of these techniques, which has long been employed in several organs facing the same issues [1,2].

This method relies on the interaction between laser light and tissues, aiming at obtaining the desired conversion of light energy into heat inside the target. This minimally invasive procedure has several advantages such as less pain and short recovery time, and it is a potential alternative to traditional resection surgery for tumor removal [2].

To address the mentioned challenge, the method of nanoparticles (NPs)-based localized increment of radiosensitization in cancerous cells is evolving as a desirable modality for enhancing radiotherapeutic ratio [3,4,5].

Many nanomaterials exhibit strong absorption in the near-infrared (NIR) range and can thus act as effective photothermal transducers. Gold nanoparticles (GNPs) are known to be effective agents for photothermal therapy (PTT). The interdisciplinary method of GNPs-mediated PTT is a promising minimally invasive thermal therapy for the treatment of focal malignancies. In this scenario, rod-shaped GNPs, i.e., gold nanorods (GNRs) have been so far proposed as the best laser photoabsorbers, since they can strongly absorb electromagnetic waves at different frequencies corresponding to the NIR radiation and can rapidly increase the temperature in the host tissue [6,7]. 

However, there are still challenges and open questions related to the long-term behavior of these NPs within the biological tissues and associated cytotoxicity, their optimal delivery approach into the tumor site, their biological distribution, and off-target effects [8,9]. All these aspects have limited the widespread transitioning of this combined technique to the clinical stages [10,11,12]. 

The failure of clinical translation of NP-mediated PTT is mainly attributed to body persistence concerns [13]. In this regard, other biodegradable nanomaterials with the ability of absorbing infrared light and converting it into heat have been designed for highly efficient photothermal therapy [14,15]. 

In order to meet this need, a family of biodegradable nano-architectures (NAs) designed within the ultrasmall-in-nano approach has been recently introduced [16]. NAs are composed by polymeric aggregates of ultrasmall NPs enclosed in approximately 100 nm silica nano capsules [17]. Besides the elimination of the building blocks through the renal pathway, NAs can take advantage of both the modifiable silica surface to enable targeted delivery, and the functionalization of the inner cavity with moieties of interest [18,19,20]. NAs have been evaluated for the mono/multi-modal chemo-photothermal treatment of HNSCCs [15,21,22]. With only a subtle modification to the original protocol for NAs, a biodegradable and excretable narrow-NIR-responsive ultrasmall-in-nano architectures (thermo-NAs, tNAs) with high photothermal capability are produced. tNAs that is part of the NAs family are non-persistent plasmonic nanomaterials designed within the ultrasmall-in-nano approach. They are composed by biodegradable silica nano-capsules containing gold ultrasmall nanoparticles (USNPs) embedded in a polymeric matrix [13,21]. The PT efficiency of tNAs has been fully assessed on a 3D customized model of human pancreatic ductal adenocarcinoma (PDAC) [15].

Porous silicon (PSi) NPs have shown attractive features for biomedical applications. Firstly, PSi is biodegradable and has good biocompatibility, because its degradation product, i.e., orthosilicic acid, is nontoxic and essential for bone and collagen growth [23]. Secondly, the highly porous framework of PSi makes it a promising platform for the delivery of both hydrophobic and hydrophilic drugs [24,25]. In addition, the recently developed black PSi (BPSi) has superior photothermal conversion performance, as compared to the widely used nanomaterials such as carbon nanotubes [14]. Thus, the BPSi NPs have been successfully used for photothermal therapy and photoacoustic tomography [14,26,27].

In the thermal therapy, the output of treatment is strictly related to the temperature distribution within the tissue. Temperature distribution is influenced by the thermal dose delivery method in the tissue and is related to the intrinsic physical properties of the tissue such as thermal properties [28]. Thermal properties vary according to temperature as a reason of thermal-induced structural modifications occurring during the treatment [29]. For designing an accurate pre-planning treatment and providing a level of therapeutic temperature, it is necessary to have the information about the tissue thermal properties as a function of temperature which is needed to understand the thermal behavior and to accurately predict the thermal outcome. 

Many mathematical models have been proposed to simulate the tissue temperature distribution in the tumor and surrounding healthy parts for treatment pre-planning with the aim of increasing the treatment efficacy and safety [30,31,32]. 

Pennes’ equation is the most commonly used formulation for biological systems to describe the heat transfer within the tissue [33].
(1)$$\rho c\frac{\partial T}{\partial t}=K\left(\nabla^{2}T\right)-Q_{b}+Q_{source}$$
in which, *K* (W/m∙K), ρ (kg/m^3^) and *c* (J/kg∙K) are thermal conductivity, density, and the heat capacity of tissue, respectively. The parameter *K* measures the ability of a material to conduct heat and can be defined by considering the heat flow through unit cross-sectional area and unit temperature gradient. The parameter *c* determines the amount of heat required by one unit of mass of the target organ to produce a unit change in its temperature. Tissue thermal diffusivity (α) is defined by the term (*K*/ρc) and measures the rate of heat transfer in a material from the hot end to the cold end [34]. In fact, thermophysical behavior of tissues in the heat transfer phenomenon can be described by this term. The term *Q_b_* refers to volumetric heat dissipation due to blood perfusion and *Q_source_* represents the energy deposition by any external heat source such as the metabolic heat production in tissues. All these thermal properties determined the ability of tissue to conduct, transfer, store, and release heat. 

In some studies, on the prediction of the ablation size during thermal treatment, thermal parameters were considered as constant values measured at room temperature (rT) [35,36]. In the Monte Carlo-based study by Cuplov et al., the effects of NPs on cancer thermal therapy were also investigated by considering the constant value for thermal properties of tissue [37]. However, considering these properties to be constant in the predictive modeling of cancer thermal treatment is not correct [38,39]. Indeed, some studies have shown the significant impact of precise definition of thermophysical properties on the more accurate modeling [40]. Nowadays, information on thermal properties of different tissues as function of temperature are available [41,42,43,44]. However, there is no information conducted to the mentioned thermal parameters for NPs-loaded organs. For the NPs-mediated combination thermal therapy, pre-treatment planning also requires knowing the behavior of thermal properties of organs when loaded with NPs.

Often, tissue-mimicking phantoms are used to assess the performances of hyperthermia simulation tools and devices, for the development of clinical methods, and for training. These phantoms usually require constituent materials whose physical properties accurately represent key tissue characteristics such as the capacity to absorb and transfer the heat [45]. In particular, agar-based phantoms have been selected thanks to their thermal properties similar to the ones of biological tissue [46,47,48,49,50]. The main advantage of using tissue-mimicking phantoms is that tissue models can be constructed with well-defined physical properties and dimensions. Moreover, when it is well characterized, the phantom allows us to compare the experimental results with model predictions.

Here, for the first time, we evaluated thermal properties and compared thermal behavior as well as laser-induced temperature distribution for three different NPs with the ability to absorb infrared light and convert it into heat. To this end, solid agar-based phantom was used to measure three thermal parameters of conductivity, diffusivity, and heat capacity for three different NPs of GNRs, BPSi NPs and tNAs in the temperature range of 25 °C to 63 °C. Additionally, temperature distribution (thermal response) was investigated and compared for these three NPs within the water phantom when it exposed to 808 nm laser beam for 5 min. 

## 2. Materials and Methods

### 2.1. Nanoparticles

To evaluate thermal behavior of GNRs we used 11-mercaptoundecyltrimethylammonnium bromide (MUTAB)-coated GNRs (supplied by the manufacturer, Nanopartz™, Inc., Loveland, Canada) with the peak absorption at 808 nm (Figure 1). The diameter (D), length (L), aspect ratio (AR) and zeta potential (ζ) of these NRs are 41 nm, 10 nm, 4.1 and 41 mV, respectively. 

BPSi NPs were prepared according to Xu et al.’s method [14]. These NPs possess a mean diameter of 156 nm and the pore diameter around 9.5 nm, and the crystallite size of 17 nm. Passion-fruit-like nano-architecture of tNAs, which are hollow silica nanocapsules with diameter of 124 nm and 20 nm thickness embedding arrays of 3 nm GNPs tNAs, were produced and characterized according to the standard procedures [15]. BPSi and tNAs were precipitated from stock solution (EtOH) by using 5 min centrifuge (13,000 rpm) and 15 min centrifuge (15,000 rpm), respectively. After drying, NPs were weighted then suspended in distilled water to make samples with concentration of 0.1 mg/mL. Before irradiation, the NPs were sonicated for a couple of minutes by using an ultrasonicated bath. The absorption spectra in a wavelength range of 300–1200 nm, transmission electron microscope (TEM) images and the size distribution for tNAs, BPSi (irregular shape) and the GNRs, are shown in Figure 1. GNRs have the maximum absorption in the wavelength around 808 nm. Although there is not any peak in the absorption spectra of the other two NPs, geometry and materials are the effective factors in the photothermal conversion.

### 2.2. Water Phantom

Agar in different concentrations is widely used for fabricating tissue phantoms [46,49]. For a concentration range of agar from 2.5% to 3.5%, the thermal properties of the agar-based phantom are similar to the thermal properties of many soft tissues [46]. In this condition, when the temperature reaches around 60 °C to 62 °C, the phase change from solid to liquid occurs. Therefore, in this work and to evaluate thermal properties of the NP-loaded phantom, thermal parameters were measured until 60 °C. 

However, since the higher temperature is needed to investigate the radiation response of NPs, to study the photothermal response of laser irradiation-exposed NP, water phantom was used instead. Laser-induced temperature distribution was investigated within the water phantom for three different NPs, i.e., GNR, tNA and BPSi, with a concentration of 0.1 mg/mL. All the experiments were performed in triplicate and were run in Eppendorf tubes (1.5 mL) containing 0.5 mL of suspension of NPs in distilled water. The control experiments were run for water alone (water did not receive NPs). 

### 2.3. Agar Phantom

Water-Agar gel phantom with a concentration of 2.5% (concentration which has been demonstrated to have similar thermal properties to that of soft tissues [46,49]) was prepared by dissolving powder agar in ultrapure water then heating and blending the solution with a magnetic stirrer hot plate until the agarose was completely dissolved. The heating of the material was stopped before it starting to boil as evaporation alters the percentage of agarose in gel. NPs were added to solution to make samples with concentration of 0.1 mg/mL, then were moved to a glass beaker with the volume of 50 mL. Each sample was allowed to solidify at rT. Agar without NPs was considered as control. The experiments were performed in triplicate. 

### 2.4. Laser Sources and Irradiation

Water phantom in the absence (control) and in the presence of NPs was exposed to laser irradiation with three power values (P) of 1.2, 2.6, and 4 W. Contact laser irradiation was performed by using Diode laser (LuOcean Mini 4, Lumics, Berlin, Germany) with wavelengths of 808 nm operating in a continuous-wave regime. A quartz optical fiber of 300 mm core diameter was used. The laser power was measured by Optical Power Meter (843-R-USB, Newport, Irvine, CA, USA). Figure 2a represents the set up used to evaluate the radiation response of water phantom with and without NPs. 

### 2.5. Temperature Monitoring System

IR thermographic camera (FLIR System, A655sc, with 640 × 480 pixels spatial resolution, ±2 °C accuracy) was used for non-contact real-time monitoring of heat distribution profile and to measure the temperature on the surface. Temperature measurements were collected for each sample exposed to laser irradiation and registered throughout the whole procedure. In each thermal image, a 30 pixel-diameter region of interest (ROI) was defined, located within the target area, encompassing the maximum temperature values. The maximum temperature value was calculated for each defined ROI.

One Fiber Bragg grating (FBG) array sensor was used to measure the temperature close to the laser applicator (2 mm distance) inside the sample irradiating with laser beam. Custom-made highly dense FBG arrays were fabricated with femtosecond point-by-point writing technology [51]. The array had 40 FBGs, each grating’s length was 1.19 mm, the distance between adjacent gratings was 0.01 mm, and the resulting length of the FBG array was 48 mm. These FBG properties were optimized to have a narrow Bragg spectral width and, at the same time, allowing accurate measurement of high-gradient temperature profiles. The temperature sensitivity of the arrays is (7.43 ± 0.01) × 10^−6^ °C^−1^ [52]. Micron Optics si255 optical interrogator (Micron Optics, Atlanta, GA, USA) was utilized to measure reflected spectra from FBG array with a 100 Hz sampling rate.

LabVIEW-based program was developed by our team to monitor temperature profile along the FBG array in real time and automatically stop laser irradiation when the maximum measured temperature reaches 100 °C. Afterwards, all data were saved and analyzed in MATLAB. In case of temperature increase up to 100 °C (measured by FBG sensor) or boiling of the solution, the irradiation was stopped.

### 2.6. Thermal Properties Measurement System

A commercial analyzer (TEMPOS, Meter Group, Inc., Pullman, WA, USA, accuracy: 10%) was used to measure thermal properties. TEMPOS has different sensors designed to measure thermal properties in liquid, semi-solid and solid materials. In this work we used the dual-needle sensor (SH-3) with 6 mm distance between the needles; each one is 30 mm long, has a diameter of 1.3 mm and has been recently approved to measure thermal conductivity, volumetric heat capacity (*C_v_*) and thermal diffusivity in solid materials [42,53]. Details regarding the mechanism of this sensor are available [50]. Detailed explanations about the algorithm that the devise uses to determine thermal conductivity and thermal diffusivity are provided in [42] and in our last paper [44]. Volumetric heat capacity is derived by having these two parameters. 

Glass beakers containing the samples were covered with a plastic bag that had two holes for insertion of the sensors into the middle part of the sample. For each measurement the beaker was placed inside the oven while it was connected to the TEMPOS analyzer that was placed outside the oven. 

Measurements were conducted on each sample over a day with a time interval of at least 60 min in order to reach equilibrium temperature and to investigate the possible changes which may be occurred in thermal properties with time and, consequently, with temperature increment. At the beginning and at the end of measurement conducted on each sample, the sensors performances were validated using the dedicated tools supplied by the manufacturer. Measurements were made over the temperature range of rT—62 °C. Figure 2b shows the schematic view of experimental setup used to measure thermal properties of the tissue mimicking phantom in the presence and the absence of NPs.

#### Measurement Uncertainty and Modeling

To clarify the quality of experimental data, uncertainty of results was measured following the guidelines of the “Guide to the expression of uncertainty in measurement” [54], and the quantities were reported for three mentioned parameters, by using the Equation (2), whose detail are available in our last paper [44].
(2)$$y_{\mathrm{Ts}}=\bar{y_{\mathrm{Ts}}}\;\pm u=\bar{y_{\mathrm{Ts}}\;}\pm k_{f}\times s$$

In this equation, *y* is the single thermal property and $\bar{y\;}$ is the arithmetic mean of the *n* = 3 measurements. The parameter $u=k_{f}\times s$ represents the expanded uncertainty of measurement, while *s* is:(3)$$s=\sqrt{\frac{\sum_{i=1}^{n}{\left(y_{T_{s,i\;}}-\bar{y_{T_{s\;}}}\right)}^{2}}{n\left(n-1\right)}}$$
and *k_f_* (coverage factor obtained from Student’s *t*-distribution, with a confidence level of 95%) is 4.30 since the experiment repeated 3 times (the degree of freedom is 2). In the temperature range of rT to 62 °C, thermal properties of the agar change almost linearly with temperature. Thus, the modeling of the behavior of this phantom follows the linear Equation (4)
*X* = A*T* + B(4)
where in *X* is thermal parameter (*K*, *D* and *C_v_*) with the slope of A and X-intercept of B, and *T* is the agar temperature. Data are presented as the mean ± standard deviation (SD). The statistical significance differences between the NPs-loaded phantoms and the control were analyzed by a *t*-test (one-tail test), to compare slopes of regression lines. The differences between groups were considered statistically significant at a *p*-value lower than 0.05.

## 3. Results

### 3.1. Irradiation and Temperature Measurement 

Figure 3 represents the highest temperature measured by FBG sensor (a) and thermal camera (b) as a function of time, respectively. As it can be seen, thermographic camera measured slightly lower temperature values than the FBG sensor. This can be explained by closer position of the FBG measurements to the laser applicator (2 mm distance) in comparison to IR measurements (Eppendorf walls). 

Among the three nanoparticles and compared to the control, GNRs showed the highest absorption of energy for all three powers. Laser irradiation with the power of 1.2 W increased temperature of GNRs to the boiling point in less than 100 s, while the same irradiation increased temperature of tNAs and BPSi NPs to 95 °C and 85 °C, respectively, in 300 s.

When the power increased to 2.6 and 4 W, the time taken by the temperature to reach the boiling points for GNRs, tNAs and BPSi decreased by around 2.8, 4 and 2.5 times and 4.65, 6, and 4 times, respectively. In this regard the speed of temperature increase was analyzed for all the NPs and for all three powers via calculation temperature gradient slope for the initial 15 s after the irradiation started. 

Figure 4 reports temporal temperature gradient obtained by FBG sensor and thermographic camera. Among the three NPs and for all the power values, the fastest temperature increase corresponded to GNRs and the lowest one corresponded to BPSi NP compared to the water sample (control). 

### 3.2. Measurement of Thermal Parameters

As was mentioned previously, a phase change from solid to semi-liquid phase was observed at around 63 °C for agar, so thermal parameters of tissue mimicking phantom was measured by increasing the temperature beginning from rT until 62 °C. Figure 5 shows the measured diffusivity (Figure 5a), conductivity (Figure 5b), and heat capacity (Figure 5c). Associated measured uncertainty is reported in the Table 1. 

The thermal parameters were almost constant for the control and tNAs NP. Diffusivity of GNRs and BPSi increased by 22% at 62 °C from the minimum level at rT. Conductivity of BPSi also increased by 20% at rT. The result indicates the significant change for conductivity of GNRs which was about 49% at 62 °C compared to the minimum value at rT. This is while this parameter was around 16% lower than that of the control in rT and the difference decreased by enhancing the temperature to around 45–50 °C. After this temperature conductivity of GNRs showed a higher value than the control in a certain temperature. These results showed the thermal effect of the GNRs as the temperature increases within the target. At 62 °C the heat capacity of all samples did not significantly change. Fitting curves associated to three thermal parameters and the measured uncertainty with a 95% confidence level with the linear fitting coefficient of A and B are represented in the Figure 6 and Table 1, respectively.

## 4. Discussion

Compared to the control, the results demonstrated the ability of all the three aforementioned NPs to convert the absorbed NIR radiation to heat. Among them, GNRs showed the highest absorption of NIR, as we could expect by considering the maximum absorbance which appears around 808 nm (Figure 1). In this figure and for the other two NPs, maximum absorption cannot be seen close to this wavelength. However, in BPSi NPs made of silicon nanomaterials (NMs) and as for other semiconductors, heat generation is due to the nonradiative recombination of free electrons and holes and for silicon NMs dominant recombination pathways are Auger decay and defect recombination. There is also a possibility of heat generation due to the thermalization of hot carriers under irradiation of higher energy than the bandgap. Additionally, according to the architecture of tNAs, PT conversion is due to a plasmonic effect rather than to the presence of silica and polymers. 

As mentioned before, the laser irradiation was automatically stopped when the temperature reached 100 °C; in case that the boiling was observed, it was stopped manually. As the results showed and it was expected, for each sample, the higher the power of radiation, the sooner the temperature increased and reached the boiling point. In a certain power of radiation, GNRs increased temperature within a shorter radiation time. For instance, for the power of 1.2 W, irradiation of sample with tNAs and Psi NPs increased the temperature up to 96 °C and 85 °C, respectively, in 5 min, while for the same power, temperature of the GNRs-loaded sample increased up to 96 °C in less than 2 min. This means the usage of GNRs as thermal agent can decrease the required irradiation time, leading to a decrease in the treatment time. Reducing treatment time lowers the radiation dose absorbed by healthy tissues and results in reduced adverse effects [55,56,57]. However, there is still a big concern related to the long-term behavior of GNRs within the biological tissues. 

The gradient of the curve related to tNAs and BPSi is partly similar to that of for the control in the lower power, but the difference between the gradient of these samples and control increased when the power increased. For the three powers, the gradient of the curve correlated to GNRs is the highest compared to that of for the other two NPs. 

The results showed the highest temperature for GNRs for the three powers. The difference between the Psi NPs and tNAs decreased as the power increased. The irradiation time was set to 300 s; however, since the samples such as tNAs and PSi NPs induce higher temperature during lower times, especially in the higher power of laser beam, the maximum temperature change was kept close to 80 °C to avoid boiling. As the results show, for the highest power the temperature change reached close to 80 °C during less than 50 s.

Regarding the absorption coefficient of gold compared to that of the silicon, the higher induced heat by gold is expected; however, the SPR (surface plasmon resonance) condition depends on the size, shape, structure, and the dielectric properties of the metal and of the surrounding medium. Compared to the solid structures, hollow or core-shell NPs show a red-shifted band of the LSPR (localized surface plasmon resonance) wavelength [15].

Additionally, the geometry parameter and the shape of NPs affect the absorption cross section (Cabs), and the SAR (specific absorption rate) is related to Cabs. LSPRs manifest themselves as a combined effect of scattering and absorption in the optical extinction spectra. Both the absorption and scattering, along with the laser type, determine the laser-target interaction. We know that NPs with high absorption produce a large amount of SAR at the entry region (the region near the laser source), while NPs with high scattering increase the internal diffuse radiation and create a SAR even bigger than the SAR of NPs with high absorption. So, the evaluation of the behavior of these NPs against higher wavelength radiation needs the pre-calculation of these parameters. Having the absorption spectra will be useful in this regard, also the mathematical calculation of cross section and absorption coefficient can help us to predict the behavior of NPs at least in comparison mode.

When the NPs-loaded tissue is exposed to irradiation, each particle generates a certain amount of heat (Q_NPs_). Temperature increases within the target that is under the laser light and loaded with these particles, are given by the heat contributed by a number (N) of NPs, which is given by the product of absorption cross section area (Cabs) and laser fluence (I) (Q_NPs_ = Cabs × I). The total heat induced within the target is obtained by multiplication of total number of NPs and the Q_NPs_ (N × Q_NPs_). The Specific Absorption Rate (SAR) for N particles is, also given by: SAR = N × Q_NPs_ = N × Cabs × I(5)
where N × Cabs defines the absorption coefficient μ_a,NPs_ of the particles. NPs with high scattering cross section increase the internal diffuse radiation and those with a high absorption cross section produce a large amount of SAR at the entry region; that is, the region near the laser source. The optical properties of the NP laden tissue are fundamental information that need to be considered and evaluated accurately and need further theoretical studies. 

Although the GNRs have shown the highest absorption of radiation and the highest conductivity for the temperature higher than 45 °C (that is in the hyperthermia range), concerns on long-term effects related to these nanoparticles still limit their clinically usage in the caner photothermal therapy. An interesting feature of tNAs that can make these NPs an ideal candidate for the photothermal agent is related to their potential employment for repeated photothermal cycles avoiding damage or re-shaping. 

The results of this study indicated the ability of all three mentioned NPs for absorbing the NIR radiation and converting to the heat and the potential of them as hyperthermia agents. The other two NPs offer some advantages including long-term body persistence, the possibility of multiple PT series (the morphology is not affected by laser irradiation, allowing for repeated PT cycles, and preserving their ability to avoid long-term body persistence in excretory system organs) and renal excretion of the building blocks after the therapeutic action, the ability of multifunctional usage for hyperthermia-assisted chemotherapy or drug delivery [14,15]. 

This work used the agar-based phantom as tissue-mimicking material for measuring thermal parameters in the presence of NPs. Compared to the average value for tissue thermal parameters, the results of this study indicated the changes of parameters when the NPs are present within the tissue. The focus of this study was on the NPs and the changes they can make to the thermal properties and irradiation response of tissue.

The use of a tissue-mimicking material holds the advantage of fabricating a homogeneous and well-defined and reproducible medium, which facilitates the embedding of NPs in a not living structure. Indeed, due to the lack of blood circulation, distribution of NPs cannot be achieved in the direct injection of particles to the organ in the ex vivo study. So, using the agar phantom in this application allows controlling the distribution of the NPs to be the most homogeneous as possible. The characteristic of homogeneous distribution is important for the measurement approach we selected for the thermal properties, as it assumes that the medium in the region of the two needles has the same properties. On the other hand, the low melting temperature of these samples did not allow us to reach temperature values typical of ablative procedures in tissues (>80 °C), but limited our analysis to about 60 °C. Additionally, the results here presented in terms of laser-induced temperature increase may vary between agar-based phantom and a biological sample: the optical properties of the agar phantom alone are not well representative of the optical response of the biological tissues, as no absorbing and scattering components are included. However, in the future, most sophisticated phantoms can be fabricated for mimicking generic tissue with well-defined absorption and scattering properties [58], in addition to the thermal properties that regulate the heat transfer.

## 5. Conclusions

Although the use of NPs as thermal agent for cancer treatment has been suggested and largely studied, limited investigation has been conducted to the thermal parameters for NPs-loaded organs. So, this study was conceived to perform an analysis of the effects of NPs on the thermal parameters. Our work firstly indicates that the presence of NPs such as GNRs, tNAs and BPSi as hyperthermia agents in treatment of tumors grant a higher temperature induced in the target and a shorter treatment time. Moreover, the results describe the change in thermal parameters in the presence of NPs and point out that thermal properties of tissue can significantly change in the presence of NPs, especially GNRs. This investigation confirms the requirement for further studies with more focus on the measurement of thermal parameters for NPs-aided hyperthermia therapy and of the irradiation response of NPs-loaded tissue, by considering the organs and the nature of NPs. This work lays the basis for a future experimental study, especially in vivo, and measurements for thermal parameters in a larger range of temperature that can provide accurate data for the clinical pre-planning of NPs-based laser therapy. 

## Figures and Tables

**Figure 1 nanomaterials-12-00945-f001:**
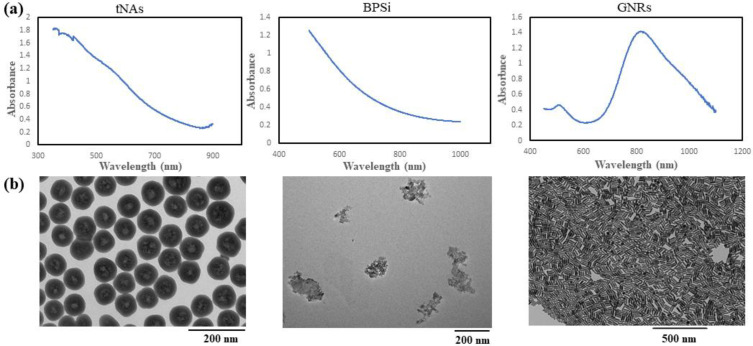
(**a**) Absorbance spectra, (**b**) TEM images and (**c**) size distribution graph for the tNAs, BPSi and GNRs.

**Figure 2 nanomaterials-12-00945-f002:**
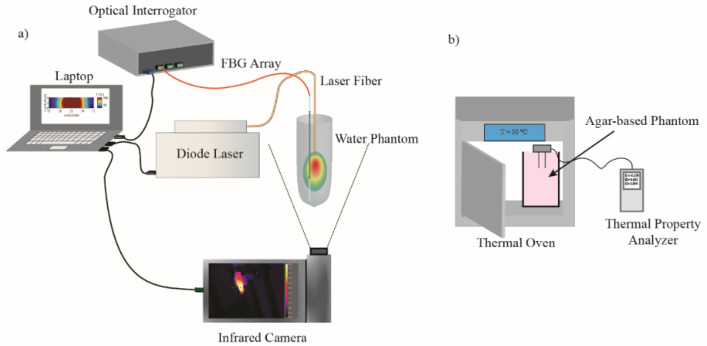
Experimental set up for (**a**) evaluation of NPs photothermal response and (**b**) measurement of NPs thermal properties.

**Figure 3 nanomaterials-12-00945-f003:**
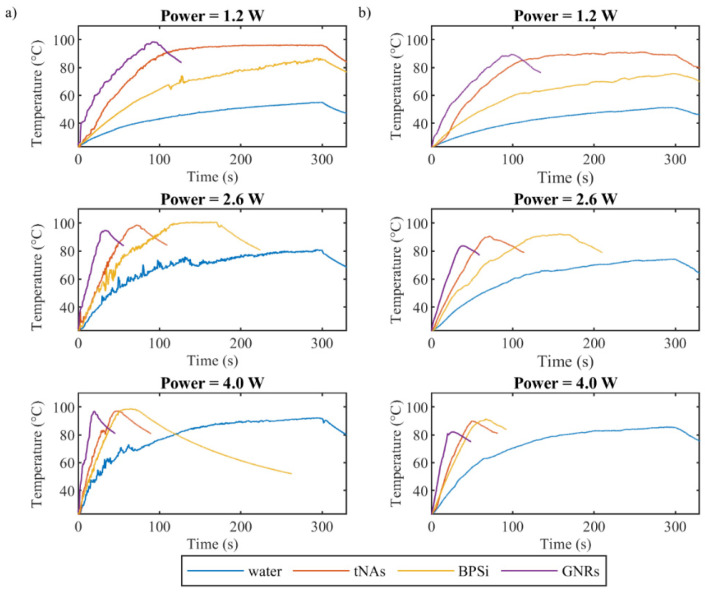
Maximum temperature in the water phantom with and without NPs measured by (**a**) FBG sensor inside the phantom and (**b**) thermal camera on the surface of phantom, during the irradiation 808 nm laser beam with three powers of 1.2, 2.6 and 4 W.

**Figure 4 nanomaterials-12-00945-f004:**
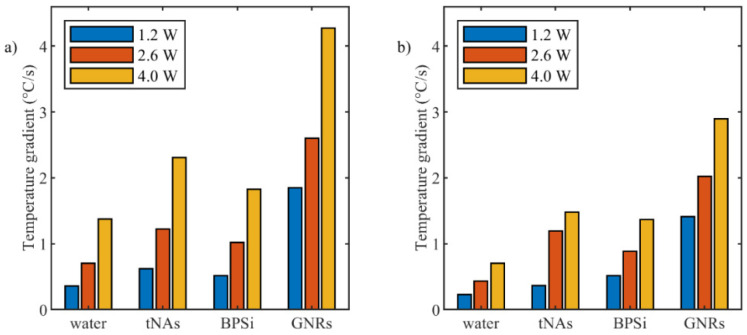
Temperature increase gradient during initial 15 s after starting of irradiation for different power levels measured by (**a**) FBG sensor and (**b**) thermographic camera.

**Figure 5 nanomaterials-12-00945-f005:**
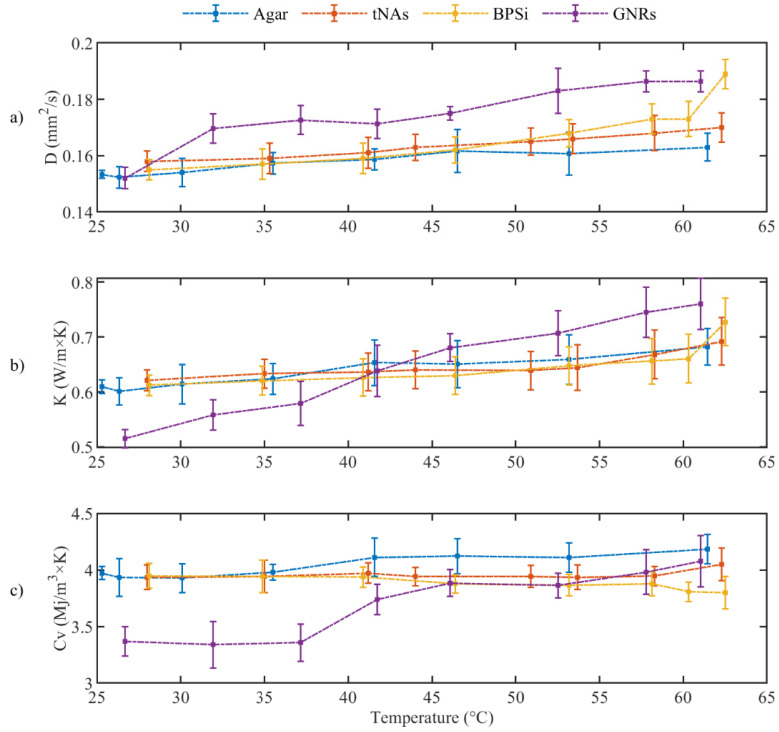
(**a**) Thermal diffusivity, (**b**) Thermal Conductivity and (**c**) Volumetric heat capacity for tissue mimicking phantom during the heating (*p*-value < 0.05).

**Figure 6 nanomaterials-12-00945-f006:**
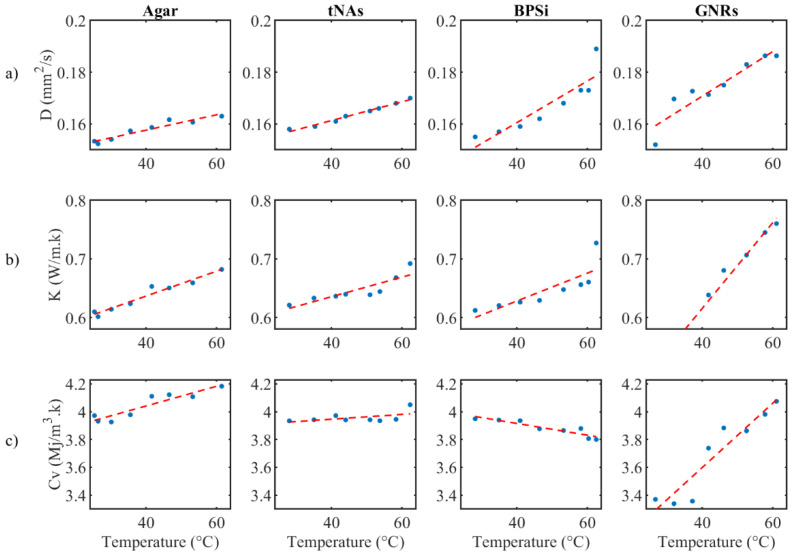
(**a**) Fitting curve for (**a**) thermal diffusivity, (**b**) thermal conductivity and (**c**) volumetric heat capacity, for agar with and without NPs.

**Table 1 nanomaterials-12-00945-t001:** Measured uncertainty associated to three thermal parameters with a 95% confidence level and the linear fitting coefficient of A and B. The 95% confidence bound is indicated within brackets.

	Coefficients	D (mm^2^/s)	K (W/m × k)	C (MJ/m^3^ × k)
Agar	A	0.0003 [0.0002, 0.0004]	0.0021 [0.00164, 0.0025]	0.0070 [0.0042, 0.0099]
	B	0.146 [0.142, 0.149]	0.551 [0.531, 0.571]	3.758 [3.636, 3.878]
Agar & tNAs	A	0.0004 [0.0002, 0.0006]	0.0019 [0.0008, 0.003]	0.0017 [0.0008, 0.0025]
	B	0.146 [0.131, 0.161]	0.595 [0.492, 0.699]	3.878 [3.741, 4.014]
Agar & PSi-NPs	A	0.0008 [0.0004, 0.0012]	0.0023 [0.0007, 0.0040]	−0.0042 [−0.0060, −0.0024]
	B	0.128 [0.109, 0.148]	0.533 [0.451, 0.6155]	4.085 [3.996, 4.173]
Agar & GNRs	A	0.0008 [0.0005, 0.0012]	0.0073 [0.0064, 0.0082]	0.0231 [0.0146, 0.0315]
	B	0.136 [0.121, 0.151]	0.323 [0.2818, 0.365]	2.676 [2.29, 3.062]

## Data Availability

The data presented in this study are available on request from the corresponding author.

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
