# Peer review of "Experimental Evaluation of Radiation Response and Thermal Properties of NPs-Loaded Tissues-Mimicking Phantoms"

_nanomaterials, 2022, doi:10.3390/nano12060945_

Round 1
Reviewer 1 Report
Paper of very good quality, but the properties measured with phantoms might not be readily applied to practical cases involving tissues.
Author Response
We really appreciate the referee understanding and his/her positive assessments of our work.
Manuscript # nanomaterials-1596517
Reviewer’s Comments and Replies
Reviewer 1
Paper of very good quality, but the properties measured with phantoms might not be readily applied to practical cases involving tissues.
Reply:
Reply: We thank the referee for the general good comment on our paper and this useful comment that caused we made the purpose and message of this work more explicit. According to this comment we revised the conclusion by adding some more explanations like what we write bellow. This revised part is highlighted in yellow color.
We agree that the data for three parameters of conduction, diffusivity and volumetric heat capacity reported in this paper cannot be considered as a precise data in the clinically planning for cancer hyperthermia, because this study was done on tissue-mimicking phantom and some parameters may be changed in the real tissue. However, as we mentioned in the manuscript, this work is a preliminary study on the effects of NPs on thermal parameters. As it is mentioned in the introduction, these parameters and the thermal behavior is needed for the pre-planning of clinically thermal therapy of cancers. In some studies, thermal parameters were considered as constant values measured at room temperature. The results indicated the change in thermal parameters in the presence of NPs. So, this study confirms the requirement for further precise study with more focus on the measurement of thermal parameters for NPs-aided hyperthermia therapy and by considering the organs and kind of NPs.

Reviewer 2 Report
In the field of hyperthermia therapy a huge number of articles are published. The submitted manuscript provides a lot of interesting information with dense discussion and suggestions. I have nothing to complain about experiments and analysis. Only a few details:
- In the Abstract (Line 17) the abbreviations for nanoparticles are given. Except of this part the "Gs" abbreviation is no longer used in the manuscript.
- NIR should be explain in the line 45 for the first time and not at the line 51.
- Figure 1 - TEM scales are low visible.
Author Response
We really appreciate the referee understanding and his/her positive assessments of our work.
Manuscript # nanomaterials-1596517
Reviewer’s Comments and Replies
Reviewer 2
In the field of hyperthermia therapy, a huge number of articles are published. The submitted manuscript provides a lot of interesting information with dense discussion and suggestions.
Reply: We appreciate the reviewer's positive feedback and are happy to address his/her comments. We have worked to improve the paper following her/his suggestions and comments.
In the Abstract (Line 17) the abbreviations for nanoparticles are given. Except of this part the "Gs" abbreviation is no longer used in the manuscript.
Reply: We really thank the Reviewer for carefully reading the paper and pointing us to this point. Gs is GNRs (gold nanorods), and a mistake happened in typing. We made it correct and highlighted in yellow
NIR should be explain in the line 45 for the first time and not at the line 51.
Reply: We revised this part and thank the referee for this right remark.
Figure 1 - TEM scales are low visible.
Reply: Thanks to the referee for this proper comment. We replaced the figure with more visible one having the scale bar.

Reviewer 3 Report
This manuscript entitled “Experimental evaluation of radiation response and thermal properties of NPs-loaded tissues-mimicking phantoms” reported a thorough characterization, thermal parameters measurements, and temperature measurements for solid phantom based on agar. The manuscript has been well written and organized. It is suitable for publication in “Nanomaterials” after revision. The authors should address the following comments
- Please confirm the abbreviations. In the abstract “Gs” should be GNRs.
- The authors should discuss some more information about conductivity and thermal capacity in the Introduction part.
- In the last part of Introduction, “diffusivity, and 129 heat capacity for three different NPs of GNRs, BPSi NPs and gold tNAs in the temperature …”, what exactly “gold tNAs” is? Also in the “Materials and Methods” should clearly the samples.
- In Figure 1, what’s the composition of “tNAs”?
- In section 2.1, please write the materials and methods used in the experiments. The author should change the content from section 2.1 to the results and discussion part.
- The concertation of NPs used was mentioned “concentration of 0.1 mg/ml”, however, how much volume was used?
- Please draw a clear scale bar to Figure 1b.
- In the manuscript, the author mentioned about particle size of nanoparticles. Please attach the particle size distribution graph in the result section.
- For temperature increase gradient, please add the thermographic camera images in Figure 4.
- In Figure 5, please elaborate on the significant change for the conductivity of GNRs and also provide information on which statistical analysis test was performed.
- As demonstrated in the discussion part, please add a specific absorption rate (SAR).
- In the discussion part, the author discussed the thermal properties of nanoparticles. Please discuss the mechanisms of NPs in tissue by absorbing radiation.
- Please discuss more about the difference and disadvantage of this simulation to the real situation.
- The author should improve the English writing.
Author Response
We appreciate the reviewer for his/her thoughtful evaluation of this manuscript and constructive comments
Manuscript # nanomaterials-1596517
Reviewer’s Comments and Replies
Reviewer 1
- Please confirm the abbreviations. In the abstract “Gs” should be GNRs..
Reply: We thank the Reviewer for carefully reading the paper and pointing us to this point. Gs is GNRs (gold nanorods), and a mistake happened in typing. We made it correct and highlighted in yellow
- The authors should discuss some more information about conductivity and thermal capacity in the Introduction part.
Reply: Thanks to referee for this comment. We have added a paragraph in the introduction and in the part that we discussed about Pennes equation and in this paragraph, we explained more about two parameters of conductivity and thermal heat capacity. One reference was added to this part. This paragraph is highlighted. Line 102-105.
- In the last part of Introduction, “diffusivity, and 129 heat capacity for three different NPs of GNRs, BPSi NPs and gold tNAs in the temperature …”, what exactly “gold tNAs” is? Also in the “Materials and Methods” should clearly the samples.
Reply: We thank the referee for this useful comment making the paper more understandable. Since tNAs is a silica shell containing gold nanosphere, we wrote Gold tNAs. But the sample named tNAs and is the abbreviation of (thermo nano-architecture). So, it was tNAs not gold tNAs and we corrected it in the last part of introduction also revised the 7th paragraph of introduction (line 70-76) and added more clear explanation about tNAs to better introducing this sample. The information about the size and geometry of this sample is also explained in “material and method” section line 150-152 (highlighted).
- In Figure 1, what’s the composition of “tNAs”?
Reply: We revised the paper and add the information about the composition and geometry of tNAs in the highlighted part (line 70-76) and line 150-152.
- In section 2.1, please write the materials and methods used in the experiments. The author should change the content from section 2.1 to the results and discussion part.
Reply: Thank the referee for reading our paper carefully. The synthesis and characterization of these NPs were done before and reported in the authors previous papers. The focus of this work is on investigating the temperature- dependency thermal behaviour of tissue-mimicking phantom in the presence of thermal-agent NPs. Since the results of this work is not related to the finding about the synthetized NPs, in the material and method, we wrote the information about these three NPs that we used in this study and referenced to the paper in where the information about synthesize procedure of them were completely reported.
As main results of this study, we reported the data about the radiation-response of these NPs-loaded tissue against laser beam and the calculated thermal parameters related to tissue-mimicking phantom in the presence and absence of NPs.
For the mentioned reasons, we prefer to keep the structure of the manuscript as it was in the submitted version.
- The concertation of NPs used was mentioned “concentration of 0.1 mg/ml”, however, how much volume was used?
Reply – For the thermal response evaluation part, 0.5 mL sample with the concentration of 0.1 mg/mL NPs suspended in distilled water were used. We highlighted this explanation in the line 180-182. For thermal parameters calculation, we used 50 ml water-agar gel tissue-mimicking phantom with the 0.1 mg/ml NPs suspended inside. This is highlighted in line 191.
- Please draw a clear scale bar to Figure 1b.
Reply: We agree with the referee and thanks her/him for this notification. We revised the figure and put the scale bar.
- In the manuscript, the author mentioned about particle size of nanoparticles. Please attach the particle size distribution graph in the result section.
Reply: We are thankful for this comment. The revision was done, and we added the size distribution for all three samples in the figure 1.
- For temperature increase gradient, please add the thermographic camera images in Figure 4.
Reply: Thanks to the referee for this suggestion. To clearly represent a temperature increasing gradient, we would need to show the movie recorded by thermal camera during the time (from the starting irradiation time to the laser off time). However, a single thermographic image (in a certain time) would show only the temperature of the sample at the same time. For this reason, we selected the maximum value of temperature inside the ROI (region of interest) of a thermal image at 15 s after turning the laser-on, and we calculated the ratio between the temperature and the selected time. We consider that this is the clearest approach for showing the temperature increase gradient.
- In Figure 5, please elaborate on the significant change for the conductivity of GNRs and also provide information on which statistical analysis test was performed.
Reply: We thanks the referee for this useful notification and advice. In this regard and in the results section, we added a paragraph to the part related to figure 5.
Also, in the last part of Material and Method Section, we added a paragraph to explain how the statistical analysis was done. Also, we add information about p-value in the figure 5. T test was used to analyse the difference between groups.
- As demonstrated in the discussion part, please add a specific absorption rate (SAR).
Reply – Thank the referee for this comment.
We revised the paper to better clarify the main purpose of this paper. In the last part of introduction and in the conclusion, we clarify that this study is a preliminary investigation on thermal behaviour change in the presence and absence of NPs. As the thermal parameters are generally considered constant in the preplanning of experimental hyperthermia studies and no study has evaluate this issue for the NPs that are design for enhance the thermal effects, we used the water and agar phantom to evaluate if the presence of NPs affect on heat induced and distribution within a target and compare three NPs based on that.
To measure the SAR we need to do theoretically evaluation of something as we explain in line 382 and 386-399:
When the NPs-loaded tissue exposed to irradiation, each particle generating a certain amount of heat (QNPs), Genarally, in presence of NPs under laser light, the temperature increase within the target is given by the heat contributed by a large number (N) of NPs, each particle generating a certain amount of heat (QNPs), which is given by the product of absorption cross section area (Cabs) and laser fluence (I) = Cabs.I). The resulting Specific Absorption Rate (SAR) for N particles is, therefore, given by:
SAR =N. QNPs= N.Cabs.I
For this preliminary study we didn’t do theoretical evaluation and consider the amount of heat induced by each NPs within the target (QNPs) and we didn’t consider and report SAR. We just compare the thermal parameters and the increased temperature induced by the three different NPs.
In the line 382-385 we explained:
So, the evaluation of the behavior of these NPs against higher wavelength radiation needs the pre-calculation of these parameters. Having the absorption spectra will be useful in this regard, also the mathematical calculation of cross section and absorption coefficient can help us to predict the behavior of NPs at least in comparison mode.
Also, we said in the conclusion, to have a precise data related to thermal parameters and the threshold of tissue for absorbing the laser in the presence of NPs, the more complete experimental studies especially in vivo and measurements for thermal parameters in a bigger range of temperature and investigation on the irradiation response of NPs-loaded tissue are needed to provide more precise and reliable data for the clinically pre-planning of NPs-based laser therapy (by considering the kind of tissue and NPs).
- In the discussion part, the author discussed the thermal properties of nanoparticles. Please discuss the mechanisms of NPs in tissue by absorbing radiation.
Reply: Thank the referee for this comment useful for improving the paper. In this regard we revised the discussion section and add more explanation in 386-399.
- Please discuss more about the difference and disadvantage of this simulation to the real situation.
Reply: Thank the referee for this suggestion. At the end of the Discussion Section, we have provided a paragraph describing the main advantages and limitations of the phantoms we have fabricated, in comparison with real biological tissues:
“The use of a tissue-mimicking material holds the advantage of fabricating a homogeneous and well-defined and reproducible medium, which facilitates the embedding of NPs in a not living structure. Indeed, due to the lack of blood circulation, distribution of NPs cannot be achieved in the direct injection of particles to the organ in the ex vivo study. So, using the agar phantom in this application allows controlling the distribution of the NPs to be the most homogeneous as possible. The characteristic of homogeneous distribution is important for the measurement approach we select for the thermal properties, as it assumes that the medium in the region of the two needles has the same properties. On the other hand, the low melting temperature of these samples did not allow to reach temperature values typical of ablative procedures in tissues (> 80 °C), but has limited our analysis to about 60 °C. Also, the results here presented in terms of laser-induced temperature increase may vary between agar-based phantom and a biological sample: the optical properties of the agar phantom alone are not well representative of the optical response of the biological tissues, as no absorbing and scattering components are included. However, in the future, most sophisticated phantoms can be fabricated for mimicking a generic tissue with well-defined absorption and scattering properties [Pogue, Brian W., and Michael S. Patterson. "Review of tissue simulating phantoms for optical spectroscopy, imaging and dosimetry." Journal of biomedical optics 11.4 (2006): 041102], in addition to the thermal properties that regulate the heat transfer.”
- The author should improve the English writing.
Reply: We Revised the paper to improve the English writing.

Round 2
Reviewer 3 Report
The authors have addressed the comments of previous reviewers. Hence, I recommend the revised manuscript for the publication in the journal of “Nanomaterials”.